

# Application of High Resolution Melt analysis (HRM) for screening haplotype variation in a non-model plant genus: Cyclopia (Honeybush)

Nicholas C. Galuszynski and Alastair J. Potts

Department of Botany, Nelson Mandela University, Port Elizabeth, Eastern Cape, South Africa

Corresponding author
Nicholas C. Galuszynski,
nicholas.galuszynski@gmail.com

## ABSTRACT

**Aim.** This study has three broad aims: to (a) develop genus-specific primers for High Resolution Melt analysis (HRM) of members of Cyclopia Vent., (b) test the haplotype discrimination of HRM compared to Sanger sequencing, and (c) provide an example of using HRM to detect novel haplotype variation in wild *C. subternata* Vogel. populations.
**Location.** The Cape Floristic Region (CFR), located along the southern Cape of South Africa.
**Methods.** Polymorphic loci were detected through a screening process of sequencing 12 non-coding chloroplast DNA segments across 14 Cyclopia species. Twelve genus-specific primer combinations were designed around variable cpDNA loci, four of which failed to amplify under PCR; the eight remaining were applied to test the specificity, sensitivity and accuracy of HRM. The three top performing HRM Primer combinations were then applied to detect novel haplotypes in wild *C. subternata* populations, and phylogeographic patterns of *C. subternata* were explored.
**Results.** We present a framework for applying HRM to non-model systems. HRM accuracy varied across the PCR products screened using the genus-specific primers developed, ranging between 56 and 100%. The nucleotide variation failing to produce distinct melt curves is discussed. The top three performing regions, having 100% specificity (i.e. different haplotypes were never grouped into the same cluster, no false negatives), were able to detect novel haplotypes in wild *C. subternata* populations with high accuracy (96%). Sensitivity below 100% (i.e. a single haplotype being clustered into multiple unique groups during HRM curve analysis, false positives) was resolved through sequence confirmation of each cluster resulting in a final accuracy of 100%. Phylogeographic analyses revealed that wild *C. subternata* populations tend to exhibit phylogeographic structuring across mountain ranges (accounting for 73.8% of genetic variation base on an AMOVA), and genetic differentiation between populations increases with distance ($p < 0.05$ for IBD analyses).
**Conclusions.** After screening for regions with high HRM clustering specificity—akin to the screening process associated with most PCR based markers—the technology was found to be a high throughput tool for detecting genetic variation in non-model plants.

## INTRODUCTION

Describing intra-population genetic diversity across a species range requires access to sufficiently variable genetic markers that can be applied to large sample sets in an efficient and cost effective manner. The lack of widely transferable marker systems with these qualities has impeded phylogeographic work in the past, especially in developing countries that harbour much of the planet's biodiversity (*Beheregaray, 2008*). High Resolution Melt analysis (HRM, sometimes acronymed to HRMA) is a high throughput and cost effective means of screening sequence variation post Polymerase Chain Reaction (PCR), offering the unique advantage of providing rapid insights into the levels of sequence variation among samples through melt curve clustering. Having the flexibility to lend itself to a variety of applications, the technology has been widely adopted in clinical (reviewed by *Vossen et al., 2009*) and crop research (reviewed by *Simko, 2016*). However, despite its apparent benefits, HRM appears to be underutilized for non-model organisms.

The HRM process is briefly described here. The inclusion of a DNA saturating fluorescent dye during PCR produces double stranded DNA molecules with dye bound to each base pair. As such, the presence of double stranded PCR product is measured by its fluorescence. As the PCR products are heated the double stranded DNA molecules dissociate, or melt, releasing the dye, resulting in a decrease in detected fluorescence. The rate at which a DNA fragment melts is dependent on the binding chemistry of the nucleotide sequence of the complementary strands under analysis. Therefore, by plotting the decrease in fluorescence against the steady rate of temperature increase, a melt curve determined by the DNA template under analysis is produced. The resultant melt curve differences (curve shape and melt peak (Tm)) are potentially indicative of sequence variation among PCR products.

The genotyping and mutation scanning abilities of HRM have been tested using well described systems in the past, including: artificially generated SNPs (*Reed & Wittwer, 2004*) and loci from the human genome (*Ebili & Ilyas, 2015*; *Garritano et al., 2009*; *Li et al., 2014*; *Reed & Wittwer, 2004*), where the technology was found to be highly sensitive and specific, with reproducible results. These studies suggest that HRM is capable of detecting single SNP variation with an average sensitivity of 95% (sd = 8%) and specificity of 97% (sd = 7%) in amplicons of various lengths (50–1,000 bp, *Reed & Wittwer, 2004*; 51–547 bp, *Li et al., 2014*; and 211–400 bp, *Garritano et al., 2009*). However, such accuracy is only possible if the starting DNA template is of sufficient quality and quantity (*Ebili & Ilyas, 2015*). Being non-destructive in nature, the PCR products can also be Sanger sequenced post HRM (*Vossen et al., 2009*). The power of the HRM approach to screen sequence variation is that it helps to avoid redundant sequencing of identical nucleotide motifs (*Dang et al., 2012*; *Vossen et al., 2009*), thereby potentially reducing overall sequencing costs of projects where intra-population genetic variation may be low, as in the slow evolving chloroplast genome of plants (*Schaal et al., 1998*). In addition, HRM has been shown to be more sensitive than traditional gel electrophoresis methods for microsatellite genotyping (*Distefano et al., 2012*). Fast, reliable and cost effective—HRM appears to be an ideal molecular tool for studies that require the characterization of a large number of samples that are likely to exhibit low nucleotide variation.

Despite its apparent utility, HRM has rarely featured in phylogeographic work. *Smith, Lu & Alvarado Bremer (2010)* were some of the first to apply HRM to population genetics. By melting short amplicons (40–60 bp) that targeted known SNPs, they successfully genotyped 121 accessions from five wild swordfish (*Xiphias gladius* Bloch, Xiphiidae) populations. *Cubry et al. (2015)* were successful in applying HRM for the discrimination of four cpDNA haplotypes that corresponded with the geographic structuring of black alder (*Alnus glutinosa* (L.) Gaertn., Betulaceae), screening 154 accessions across 23 populations. These studies, and most others applying HRM to non-model organisms (*Dang et al., 2012*; *Li et al., 2012*; *Radvansky et al., 2011*), set out to develop HRM primers having prior knowledge of the nucleotide variation under analyses. Unfortunately, such knowledge is generally not available for the study of non-model organisms and the application of HRM for detecting and genotyping of novel genetic variation in wild populations is still rare (*Nunziata et al., 2019*; *Sillo et al., 2017*). High Resolution Melt analysis appears to be an underutilized resource by phylogeographers.

Here we test the application of HRM for non-model taxa, *Cyclopia*, a commercially important plant genus endemic to the CFR. This study: (a) develops a set of genus-specific primers for the HRM analysis of non-coding cpDNA loci to test: (b) the haplotype discrimination sensitivity, specificity, and accuracy of HRM, and (c) the potential application of HRM for haplotype detection in wild *Cyclopia* populations, focusing here on *C. subternata*. This study demonstrates that (when optimized) HRM is a fast, accurate, and cost effective tool for haplotype detection in non-model organisms, successfully describing the geographic structuring of genetic diversity in wild *C. subternata* populations.

## MATERIALS & METHODS

### Taxonomic background and sampling

This study focuses on members of the genus *Cyclopia* Vent., which is endemic to the Cape Floristic Region (CFR) and consists of 23 described species; two of which are considered extinct (*Cyclopia filiformis* Kies, *Cyclopia laxiflora* Benth.) and various others ranging from critically endangered to vulnerable (*SANBI, 2017*). *Cyclopia* species and populations tend to exhibit highly localised distributions (*Schutte, 1997*), making them potentially vulnerable to genetic pollution from foreign genotypes translocated for the cultivation of Honeybush tea and associated products (*Ellstrand & Elam, 1993*; *Levin, Francisco-Ortega & Jansen, 1996*; *Potts, 2017*; *Schutte, 1997*)—an increasingly common practice in the CFR (*McGregor, 2017*). The characterization and conservation of wild *Cyclopia* genetic diversity is therefore of high importance.

To maximise the amount of genetic variation detected and the transferability of the primers designed across the genus, 14 species (summarized in Table 1, closed circles in Fig. 1) were sampled from the full geographic range of the genus. Additionally, eight wild populations (open circles in Fig. 1) of *C. subternata* Vogel. were sampled to test the potential application of HRM for haplotype detection using the genus-specific primers generated. Between 10 and 24 samples were collected per *C. subternata* population. Fresh leaf material was clipped from the growing tips of wild specimens over the period of

**Table 1  Species and non-coding cpDNA regions screened for HRM primer development for the non-model plant genus *Cyclopia* Vent.**

| Species | Non-coding cpDNA regions sequenced (dependent on successful amplification) | | | | | | | | | | | |
|---|---|---|---|---|---|---|---|---|---|---|---|---|
| | *rpl32-trnL* intergenic spacer | *ndhA* intron | *trnQ-50rps16* intergenic spacer | *atpI-atpH* intergenic spacer | *petL-psbE* intergenic spacer | *trnD-psbM* intergenic spacer | *trnG-trnG2G* intergenic spacer | *30trnV-ndhC* intergenic spacer | *TrnfM-trnS* | *psbJ-petA* | *psaI-accD* | *psbD-trnT* |
| *C. alpina* | | X | | | | | | X | X | X | | |
| *C. aurescens* | | X | | X | | | | X | X | X | X | X |
| *C. bolusii* | | X | | | | | | | | | | |
| *C. burtonii* | | | X | X | | | X | X | | | | |
| *C. buxifolia* | | X | | X | X | | X | | | | | |
| *C. galioides* | | | | X | | | | | | | | |
| *C. genistoides* | | X | | X | | | X | | | X | | X |
| *C. intermedia* | X | X | X | X | | X | X | X | X | X | X | X |
| *C. longifolia* | X | | X | X | X | X | X | X | X | X | X | X |
| *C. maculata* | | X | | | | | X | | | X | X | X |
| *C. plicata* | X | X | | | | | | X | X | X | X | |
| *C. pubescens* | X | X | X | X | X | X | X | X | X | X | X | |
| *C. sessilifolia* | X | X | | X | X | X | X | | X | X | X | |
| *C. subternata* | X | X | | X | X | X | X | X | X | X | X | X |

Following PCR amplification, only clear, bright bands visualized through gel electrophoresis were selected for sequencing with a maximum of six and a minimum of one sample selected for sequencing per species.
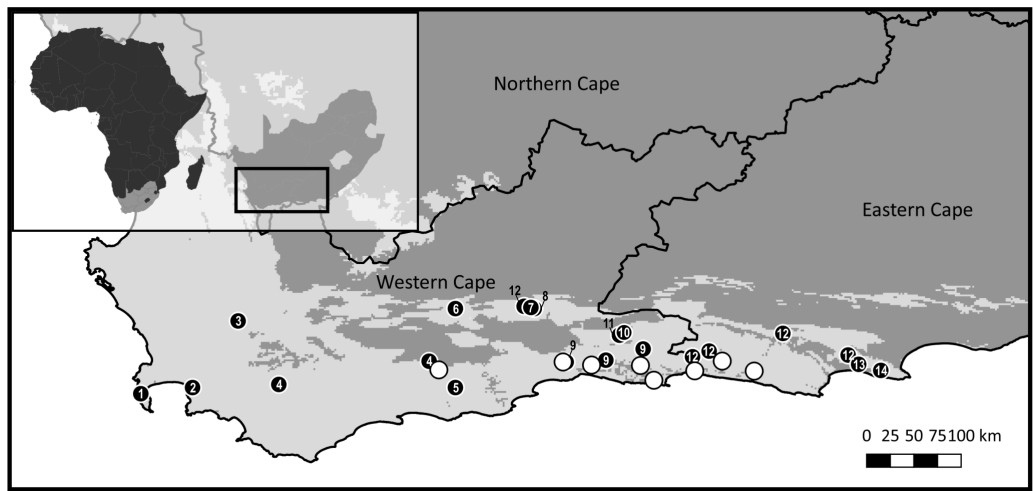

**Figure 1** **Sample distribution map.** Study domain superimposed with the distribution of the CFRs fynbos biome, to which *Cyclopia* is endemic. Inset indicates the position of the study domain in relation to South Africa and the African continent. Distribution of samples included in non-coding cpDNA haplotype screening for HRM primer development are displayed (filled circles) in conjunction with the locations of the *C. subternata* populations included in the phylogeographic analysis (open circles). Closed circles are numbered based on species identity: 1 = *C. galioides*, 2 = *C. genistoides*, 3 = *C. buxifolia*, 4 = *C. maculata*, 5 = *C. sessilifolia*, 6 = *C. burtonii*, 7 = *C. aurescens*, 8 = *C. bolusii*, 9 = *C. subternata*, 10 = *C. plicata*, 11 = *C. alpina*, 12 = *C. intermedia*, 13 = *C. longifolia*, 14 = *C. pubescens*.

2015–2018 and placed directly into a silica desiccating medium for a minimum of two weeks prior to DNA extraction. All sampling was approved by the relevant permitting agencies, Cape Nature (Permit number: CN35-28-4367), the Eastern Cape Department of Economic Development, Environmental Affairs and Tourism (Permit numbers: CRO 84/ 16CR, CRO 85/ 16CR), and the Eastern Cape Parks and Tourism Agency (Permit number: RA_0185).

## DNA extraction

Whole genomic DNA was extracted from silica-dried leaf material using a CTAB approach modified from *Doyle & Doyle (1987)*, the full extraction protocol is described in Methods S1. Extracted DNA was suspended in 50 μL molecular grade water for PCR amplification with the products sequenced using Sanger sequencing (*Sanger, Nicklen & Coulson, 1977*). Samples that failed to amplify during PCR, were subject to repeat DNA extracted from new leaf material and then PCR amplified.

## Developing *Cyclopia* specific HRM primers

While HRM has been shown to successfully detect sequence variation in PCR products of various sizes (see introduction), it has been suggested that shorter PCR products are likely to produce more pronounced melt curve differences than larger products with the same nucleotide variation (*Dang et al., 2012*; *Dobrowolski et al., 2009*; *Li et al., 2014*; *Liew et al., 2004*; Smith et al. 2013; *Taylor et al., 2011*). Universal marker systems, such as those developed by *Shaw et al. (2005)* and *Shaw et al. (2007)* are therefore unlikely to be directly

transferable to HRM, as they amplify relatively large PCR products, thus HRM specific primers must be developed to target shorter, variable regions.

Developing HRM primers requires prior knowledge of the nucleotide variation of regions across samples. The means of acquiring such data is dependent on the resources available to the researcher and the availability of existing sequence data for the study organisms. Thus template data could range from Next Generation Sequencing derived genomic data to the application of HRM to existing microsatellite markers, or existing data available from international nucleotide sequence databases such as GenBank (https://www.ncbi.nlm.nih.gov/genbank/).

For *Cyclopia*, however, existing sequence data (predominantly from the ribosomal ITS region) exhibited low levels of differentiation amongst species (*Galuszynski & Potts, 2017*; *Van Der Bank et al., 2002*), lacking the variation required for population level analyses. Therefore, polymorphic loci were identified from non-coding cpDNA regions via Sanger sequencing (*Sanger, Nicklen & Coulson, 1977*) of PCR products amplified using the protocols and universal primers described by *Shaw et al. (2005)* and *Shaw et al. (2007)*. A total of 16 non-coding cpDNA regions under went PCR, however four regions failed to amplify (and could not be sequenced). The 12 regions that were sequenced are summarized in Table 1, all necleotide sequence data is availible from GenBank and accession numbers are provided in Table S1.

Sequences were assembled using CondonCode Aligner *[v2.0.1]* (Codon Code Corp, http://www.codoncode.com). The PHRED base-calling program (*Ewing et al., 1998*) was used to assign a quality score for each sequence, then sequences were automatically aligned using ClustalW (*Thompson, Higgins & Gibson, 1994*) and visually inspected for quality. All short indels (<3 bp) occurring in homopolymer repeat regions were considered alignment errors and removed from the alignment. The consensus sequence alignment for polymorphic regions were exported and utilized in HRM primer design.

Primer design was guided by two constraining factors: (1) sequences had to contain conservative regions with a high GC content that could form the primer binding template, and (2) these regions had to flank polymorphic sites. Wherever possible, internal HRM primers were designed in a way that would split a region into neighboring loci, as suggested by *Dang et al. (2012)*. This approach allows for adjacent loci to be sequenced in a single run by amplifying the full region, and then during alignment, split the region into the neighboring loci that underwent HRM analysis. This approach reduces the time involved in sequence alignment and number of samples required to be sequenced for HRM clustering verification.

High Resolution Melt specific primers were designed using the online resource Primer-Blast (http://www.ncbi.nlm.nih.gov/tools/primer-blast/). The sub-family Faboideae was used as the reference taxon to check for primer specificity searched against the NCBI Reference Sequence representative genomes (http://www.ncbi.nlm.nih.gov/refseq/); PCR product size was limited to between 50 and 550 bp (as this falls within the amplicon size predicted to produce the highest levels of genotyping accuracy; *Dang et al., 2012*; *Dobrowolski et al., 2009*; *Li et al., 2014*; *Liew et al., 2004*; *Taylor et al., 2011*), primer melting temperature was set at 60 °C (± 3 °C) (as suggested by *Taylor et al., 2011*) and a maximum

**Table 2** *Cyclopia* **specific primers designed for testing HRM haplotype discrimination.** Primers used to screen haplotype variation in wild *C. subternata* populations are indicated in bold. Primer details provided include; non-coding cpDNA region the primers are located in, as well as each primers' annealing temperature (Tm) in degrees Celcius, GC content, and sequence motif.

| Region | Primer | Direction | Tm (°C) | GC (%) | Sequence (5′ → 3′) |
|---|---|---|---|---|---|
| *trnG* intron | MLT_C1 | F | 57.3 | 43.5 | ACTCCTCTTCTATTCATGGGGA |
| | MLT_C2 | R | 58.0 | 50.0 | |
| | MLT_C3 | F | 61.8 | 40.9 | TCAACGAACGATTCGAGGAATA |
| | MLT_C4 | R | 61.1 | 45.5 | TGCTTCAATCTCTCCTACCCAA |
| *pctL-psbE* intergenic spacer | MLT_M1 | F | 58.0 | 43.5 | TGTCGAGAACCCTTATACTCTCA |
| | MLT_M2 | R | 58.7 | 47.6 | TACCAAGGGTGTCTTTCGAGT |
| *atpI-atpH* intergenic spacer | **MLT_S1** | **F** | 64.3 | 50 | ATTACAGATGAAACGGAAGGGC |
| | **MLT_S2** | **R** | 61.5 | 45.5 | TGGGGGTTTCAAAGCAAAGG |
| | **MLT_S3** | **F** | 61.5 | 45.5 | CCTTTGCTTTGAAACCCCCA |
| | **MLT_S4** | **R** | 66.4 | 36.5 | TTCCCGTTTCATTCATTCACATTCA |
| *ndhA* intron | **MLT_U1** | **F** | 59.1 | 40.0 | AGGTACTTCTGAATTGATCTCATCC |
| | **MLT_U2** | **R** | 62.2 | 52.4 | GCAGTACTCCCCACAATTCCA |
| *rpl32-trnL* intergenic spacer | MLT_V1 | F | 59.9 | 60.0 | CTCCTTCCCTAAGAGCAGCG |
| | MLT_V2 | R | 59.2 | 40.0 | GTTGGAATAATCTGAATTAGCCGGA |

of 20 primer pairs were returned per search. The positions of these primers within their respective region alignment were manually evaluated to ensure that they occurred in well conserved sites, i.e., any primers occurring across polymorphic loci were discarded.

Eleven genus-specific primer pairs (Table S2) were developed from seven of the twelve non-coding cpDNA regions, of which eight primer pairs successfully amplified PCR products and were thus selected for HRM screening (Table 2). The remaining three were excluded from the analysis due to poor PCR amplification. The primer pairs selected for HRM screening amplified between four and six unique haplotypes each, across five cpDNA regions (nucleotide differences are summarized in Table 3). Primers selected for the evaluation of HRM accuracy are reported in Table 2.

## Testing PCR amplification of HRM primers

The genomic DNA extracted for samples that amplified unique haplotypes (as determined from the sequence data used to develop HRM primers) was quantified using a NanoDrop 2000c spectrophotometer (Thermo Fisher Scientific, Wilmington, DE19810r Scientific, USA) and 5 ng/L DNA dilutions were made for HRM analysis. High Resolution Melt analysis was conducted for all primer pairs developed, with 16 replicates amplified per sample (haplotype). However, only replicates that produced sufficient PCR products, as determined from PCR amplification curves (see examples in Figs. 2 and 3) were included in the evaluation of HRM haplotype discrimination (number of replicated subjected to HRM analysis for each haplotype are reported in Table 3). This PCR amplification screening approach was adopted as the aim of this phase of the study was to test the haplotype discrimination abilities of HRM based on the underlying nucleotide differences between haplotypes and not the quantity and quality of PCR product under analysis (which can

Galuszynski and Potts (2020), *PeerJ*, DOI 10.7717/peerj.9187

**Table 3 Nucleotide differences and clustering results for HRM discrimination of known haplotype.** Sample ID of the accessions that were PCR amplified in replicates of 16, the number of replicates that successfully amplified during PCR and subject to HRM analysis is given (N), followed by HRM haplotype discrimination (sensitivity, specificity and accuracy), the grouping of each replicate into a HRM cluster is provided for each haplotype amplified per primer combination (clusters 1–11), a summary of the nucleotide differences between haplotypes is also provided.

| Primer pair | N | Sen | Spe | Acc | HRM grouping of replicates into cluster 1–cluster 11 | | | | | | | | | | | Nucleotide difference summary | | | |
|---|---|---|---|---|---|---|---|---|---|---|---|---|---|---|---|---|---|---|---|
| | | | | | 1 | 2 | 3 | 4 | 5 | 6 | 7 | 8 | 9 | 10 | 11 | | | | |
| MLT C1-MLT C4 (150 bp) (TrnG intron) | | | | | | | | | | | | | | | | 19 | 20 | 72 | 205 |
| | | | | | | | | | | | | | | | | T | A | T | A |
| Haplotype A | 14 | 71 | 94 | 88 | 2 | 10 | | 2 | | | | | | | | G | T | G | . |
| Haplotype B | 16 | 69 | 44 | 52 | 11 | | 4 | | 1 | | | | | | | . | . | . | . |
| Haplotype C | 11 | 91 | 49 | 58 | 10 | 1 | | | | | | | | | | . | . | . | C |
| Haplotype D | 11 | 73 | 44 | 50 | 8 | 1 | 3 | | | | | | | | | . | . | G | . |
| MLT C3-MLT C4 (236 bp) (TrnG intron) | | | | | | | | | | | | | | | | 41 | 48-55 | 62 | |
| | | | | | | | | | | | | | | | | T | # | A | |
| Haplotype A | 14 | 100 | 43 | 57 | 14 | | | | | | | | | | | . | # | T | |
| Haplotype B | 16 | 56 | 27 | 36 | 9 | 6 | 1 | | | | | | | | | . | – | . | |
| Haplotype C | 12 | 58 | 79 | 75 | 4 | 7 | | 1 | | | | | | | | G | # | . | |
| Haplotype D | 14 | 79 | 36 | 46 | 11 | 3 | | | | | | | | | | . | # | . | |
| | | | | | | | | | | | | | | | | # = AAAAATTG | | | |
| MLT M1-MLT M2 (170 bp) (pctL-psbE inter-genic spacer) | | | | | | | | | | | | | | | | 84 | 88 | 110 | 118 |
| | | | | | | | | | | | | | | | | G | G | G | A |
| Haplotype A | 15 | 93 | 98 | 97 | | 14 | 1 | | | | | | | | | A | . | . | . |
| Haplotype B | 16 | 94 | 67 | 74 | 15 | | | 1 | | | | | | | | . | . | . | . |
| Haplotype C | 14 | 93 | 98 | 97 | 1 | 13 | | | | | | | | | | . | T | . | . |
| Haplotype D | 16 | 94 | 67 | 74 | 15 | | | 1 | | | | | | | | . | . | T | G |
| MLT S1-MLT S2 (217 bp) (atpI-atpH inter-genic spacer) | | | | | | | | | | | | | | | | 53 | 54 | 62-80 | 95 |
| | | | | | | | | | | | | | | | | T | A | – | G |
| Haplotype A | 12 | 100 | 100 | 100 | | | 12 | | | | | | | | | . | . | – | . |
| Haplotype B | 15 | 100 | 100 | 100 | 15 | | | | | | | | | | | . | . | – | A |
| Haplotype C | 11 | 100 | 100 | 100 | | | 11 | | | | | | | | | A | C | – | . |
| Haplotype D | 14 | 100 | 100 | 100 | | 14 | | | | | | | | | | A | C | # | . |
| | | | | | | | | | | | | | | | | # = TTCATAGATAACTAGTTAG | | | |

**MLT S1-MLT S4 (527 bp) (atpI-atpH intergenic spacer)**

| Primer pair | N | Sen | Spe | Acc | 1 | 2 | 3 | 4 | 5 | 6 | 7 | 8 | 9 | 10 | 11 | 75 | 76 | 86-104 | 117 | 267 | 281 | 287 | 382 | 477-481 |
|---|---|---|---|---|---|---|---|---|---|---|---|---|---|---|---|---|---|---|---|---|---|---|---|---|
| | | | | | | | | | | | | | | | | T | A | – | G | C | C | T | C | * |
| Haplotype A | 14 | 100 | 79 | 83 | 14 | | | | | | | | | | | . | . | – | . | T | . | . | . | – |
| Haplotype B | 10 | 80 | 100 | 98 | | | | | 8 | 2 | | | | | | . | . | – | A | . | . | . | . | – |
| Haplotype C | 14 | 100 | 79 | 83 | 14 | | | | | | | | | | | . | . | – | . | T | . | . | . | * |
| Haplotype D | 14 | 86 | 100 | 98 | | | 12 | | | | 2 | | | | | A | C | # | . | . | T | G | . | * |
| Haplotype E | 16 | 100 | 100 | 100 | | 16 | | | | | | | | | | . | . | – | A | . | . | . | A | – |
| Haplotype F | 14 | 71 | 100 | 95 | | | | 10 | | | | 2 | 2 | | | A | C | – | . | . | T | G | . | * |

# = CATAGATAACTAGTTAGTT, * = TTTTC

**MLT S3-MLT S4 (310 bp) (atpI-atpH intergenic spacer)**

| Primer pair | N | Sen | Spe | Acc | 1 | 2 | 3 | 4 | 5 | 6 | 7 | 8 | 9 | 10 | 11 | 52 | 66 | 72 | 167 | 262-266 |
|---|---|---|---|---|---|---|---|---|---|---|---|---|---|---|---|---|---|---|---|---|
| | | | | | | | | | | | | | | | | C | C | T | C | – |
| Haplotype A | 15 | 100 | 100 | 100 | | 15 | | | | | | | | | | T | . | . | . | – |
| Haplotype B | 12 | 92 | 100 | 95 | | | 11 | | | | 1 | | | | | . | . | . | . | – |
| Haplotype C | 11 | 91 | 100 | 95 | | | | | 10 | 1 | | | | | | T | . | . | . | # |
| Haplotype D | 16 | 100 | 100 | 100 | 16 | | | | | | | | | | | . | T | G | . | # |
| Haplotype E | 14 | 93 | 100 | 95 | | | 13 | | | | | 1 | | | | . | . | . | A | – |

# = TTTTC

**MLT U1-MLT U2 (345 bp) (ndhA intron)**

| Primer pair | N | Sen | Spe | Acc | 1 | 2 | 3 | 4 | 5 | 6 | 7 | 8 | 9 | 10 | 11 | 15 | 22 | 47 | 79 | 135-141 | 149 | 172 | 183 | 220 | 253 | 289 |
|---|---|---|---|---|---|---|---|---|---|---|---|---|---|---|---|---|---|---|---|---|---|---|---|---|---|---|
| | | | | | | | | | | | | | | | | T | T | G | C | # | C | A | A | G | T | A |
| Haplotype A | 16 | 100 | 100 | 100 | 16 | | | | | | | | | | | . | . | . | . | # | . | . | . | . | . | . |
| Haplotype B | 11 | 73 | 100 | 96 | | | 8 | | | 3 | | | | | | . | C | . | A | # | A | C | G | . | G | T |
| Haplotype C | 15 | 93 | 100 | 99 | | 14 | | | | | | | | | 1 | C | . | . | . | # | . | . | . | . | . | . |
| Haplotype D | 16 | 63 | 100 | 91 | | | | | 10 | 3 | | 2 | 1 | | | . | . | A | . | # | . | . | . | . | . | . |
| Haplotype E | 12 | 92 | 100 | 99 | | | | 11 | | | | | | 1 | | . | . | A | . | – | . | . | . | A | . | . |

# = TATCCCC

**MLT V1-MLT V2 (340 bp) (rpl32-trnL intergenic spacer)**

| Primer pair | N | Sen | Spe | Acc | 1 | 2 | 3 | 4 | 5 | 6 | 7 | 8 | 9 | 10 | 11 | 34-38 | 56 | 104 |
|---|---|---|---|---|---|---|---|---|---|---|---|---|---|---|---|---|---|---|
| | | | | | | | | | | | | | | | | # | T | T |
| Haplotype A | 14 | 86 | 65 | 71 | 12 | 2 | | | | | | | | | | – | . | . |
| Haplotype B | 10 | 91 | 99 | 98 | | | 10 | | | | | | | | | # | . | . |
| Haplotype C | 12 | 92 | 66 | 73 | 12 | | | | | | | | | | | # | A | A |
| Haplotype D | 10 | 100 | 89 | 92 | 10 | | | | | | | | | | | # | A | . |

# = ATTATT

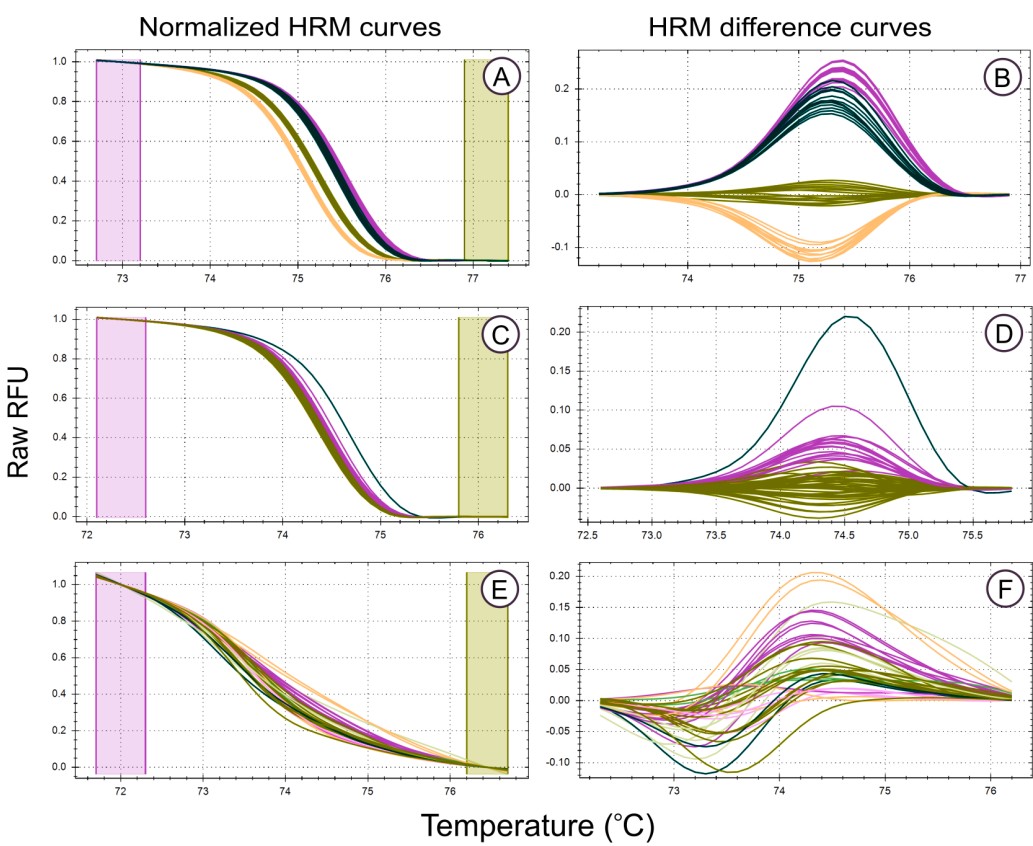

**Figure 2 High Resolution Melt curve examples.** Melt curves and their difference curves for the PCR products amplified by three of the genus specific primers developed. Curves are ordered in decreasing HRM clustering accuracy and the bottom curves (E, F) were generated using the primer pair MLT T1-MLT T2 (*TrnQ-5'rps16* intergenic spacer) that was excluded from HRM analysis due to poor amplification resulting in inconsistent melt curve production, the details of this primer pair, in addition to all primer pairs that were excluded from HRM haplotype discrimination analysis due to poor PCR amplification, are provided in Table S1. HRM curves (A, C, E), the normalized change in florescence associated with PCR product dissociation when heated. Melt domain identification and melt curve normalization was automated by the HRM software in this study, this process may be required to be performed manually on other platforms. A reference melt curve is selected and used as a baseline to plot melt curve differences across the melt domain, therefore difference curves (B, D, E) have different *X* axes. HRM clusters are automatically generated and colourised by the HRM software used. Melt curves were generated from the PCR products generated using the primer pairs, (A, B) MLT S1–MLT S2 (*atpI-atpH* intergenic spacer), (C, D) MLT C3–MLT C4 (*trnG-trnG2G* intergenic spacer), and (E,F) MLT T1–MLT T2 (*trnQ-5'rps16* intergenic spacer).

vary due to pippetting errors). Regions that failed to produce consistent PCR amplification curves (possibly due to non-specific primer binding), were excluded from subsequent analysis (see examples of PCR and HRM curves excluded from analysis in Figs. 2 and 3).

## PCR and HRM reactions

All reactions (PCR amplification and subsequent HRM) took place in a 96 well plate CFX Connect (Bio-Rad Laboratories, Hercules, California, U.S.A.) in 10 µL reaction setups, consisting of 4 µL genomic DNA (5 ng/µL), 1 µL each primer (10 mM) and 5 µL Precision
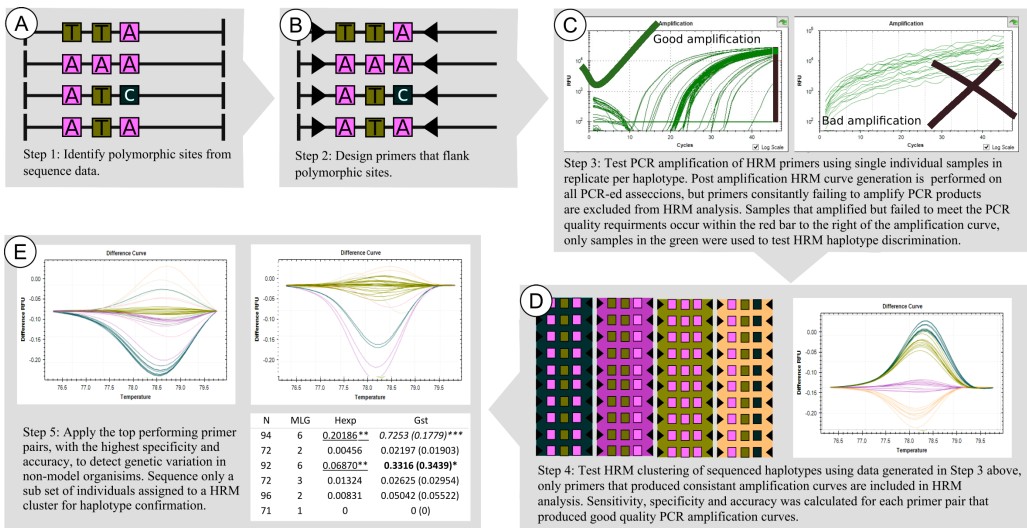

**Figure 3** **Framework used to developed, test, and apply HRM to the genus *Cyclopia*, a group of non-model organisms.** This involves identifying polymorphic loci (A), designing taxon specific primers (B), testing PCR amplification success of the taxon specific primers (C), testing the HRM clustering accuracy of PCR products of known nucleotide sequence motif (D), and then screening novel nucleotide variation across loci that have proven to result in high HRM accuracy (E).

**Table 4** **Protocol for PCR amplification and subsequent HRM curve generation.** Primer specific annealing temperatures (Tm) are provided in Table 2.

| Process | Step | Temperature | Time | Number of cycles |
|---|---|---|---|---|
| PCR Amplification | Initial Denaturing | 95 °C | 2 min | 1 |
| | Denaturing | 95 °C | 10 sec | |
| | Annealing/Extension + Plate Read | Primers mean Tm | 30 sec | 40 |
| | Extension + Plate Read | 72 °C | 30 sec | |
| HRM analysis | Heteroduplex formation | 95 °C | 30 sec | 1 |
| | | 60 °C | 1 min | 1 |
| | HRM + Plate Read | 65–95 °C (in 0.2 °C increments) | 10 sec/step | 1 |

Melt Supermix containing hot-start iTaqTM DNA polymerase, dNTPs, MgCl2, EvaGreen dye (Bio-Rad Laboratories, Hercules, California, U.S.A.).

Polymerase Chain Reaction amplification and melt conditions were as per manufacturer's specifications (Table 4) and the annealing temperature set to the primer pair's mean Tm (melting temperature), reported in Table 2. The automated clustering algorithm of the High Precision Melt software[TM] (Bio-Rad Laboratories, Hercules, California, U.S.A.) was performed on the normalized florescence data and used to group melt curves into clusters that represent putative haplotypes. HRM clustering settings used were ΔTm threshold at 0.05 °C and curve shape sensitivity settings and temperature correction, 70% and 20 respectively.

## HRM discrimination of sequenced haplotypes

Following the descriptions of *Altman and Bland (1994)*, HRM discrimination (sensitivity, specificity and accuracy) was determined for each of the haplotypes amplified by the eight HRM primers that produced sufficient PCR product for HRM analysis. Sensitivity, or the true positive rate, refers to HRM's ability to correctly assign haplotype replicates into the same HRM cluster.

$$Sensitivity = TP/(TP + FN)$$

$$TP = TruePositive \quad FN = FalseNegative$$

Specificity, or true negative rate, is the measure of HRM's ability to correctly discern between haplotypes, grouping them into different HRM clusters.

$$Specificity = TN/(TN + FP)$$

$$TN = TrueNegative \quad FP = FalsePositive$$

The accuracy of HRM refers to how close haplotype clustering reflects the true identities of the haplotypes and was measured as:

$$Accuracy = (TP + TN)/(TP + FP + TN + FN)$$

Since sensitivity below 100% will be accounted for during HRM cluster (i.e., putative haplotype) confirmation by sequencing (with a subset of samples from each unique HRM cluster sequenced), all regions with 100% specificity were included for the detection of novel haplotypes in wild *C. subternata* populations.

## The potential for HRM to detect haplotype variation in wild populations

Only three regions (MLT S1–MLT S2, MLT S3–MLT S4, and MLT U1–MLT U2) were found to have an HRM clustering specificity of 100% (Fig. 4). Thus these regions were screened for haplotype variation across 142 accessions from eight wild *C. subternata* populations.

The same approach as *Dang et al. (2012)* was employed, with each sample run in duplicate and haplotype clustering performed on a single population basis with the intention of reducing errors resulting from variation of PCR product concentration and quality across samples from different population extractions. This was achieved by using the built in well group function in the CFX Manager™ Software (Bio-Rad Laboratories, Hercules, California, U.S.A.), thus multiple populations could be included in a run, but analyses separately for HRM clustering.

The cpDNA regions that were used to design the primers used for HRM haplotype detection were amplified and sequenced (following the same protocols as before) to confirm the haplotype identity of HRM clusters. The loci amplified by MLT S1–MLT S2 and MLT S3–MLT S4 are adjacent to one another and by sequencing the full atpI-atpH intergenic spacer, the sequence identity of both loci could be confirmed with reduced sequencing and alignment effort. Moreover, the position of the loci amplified by the HRM primers occurred near the center of their respective parent regions and unidirectional

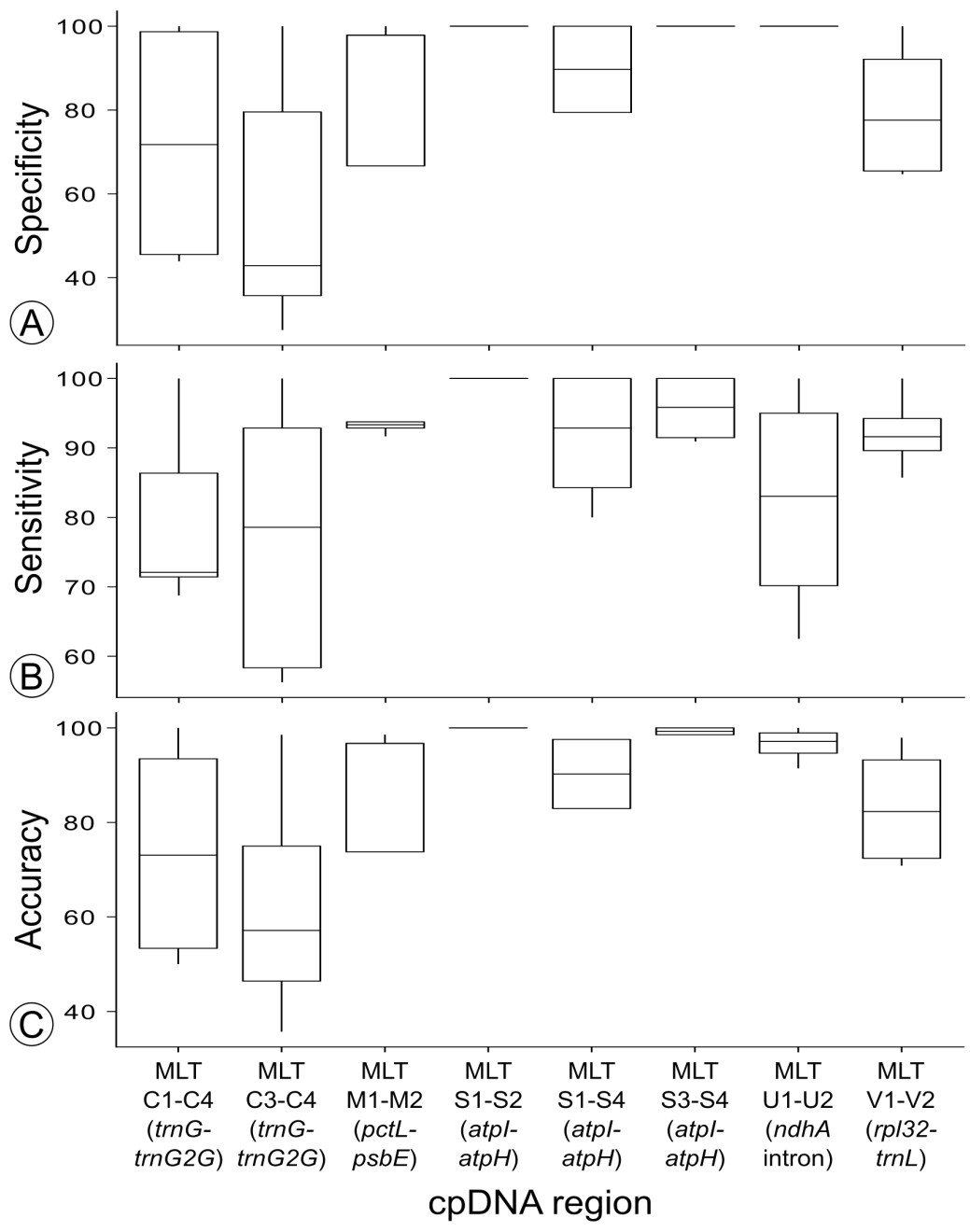

**Figure 4** Summary of the (A) specificity, (B) sensitivity and (C) accuracy for the regions used to test haplotype discrimination by HRM.

sequencing using the reverse primers of *Shaw et al. (2007)* proved sufficient for verifying the sequence motifs under HRM analysis. A minimum of three accessions representing each HRM cluster (i.e., putative haplotype) in each population were sequenced for haplotype verification. Samples whose replicates were classified as two different clusters, thus having uncertain haplotype identity, were also sequenced to ensure they were assigned correctly.
A total of 46 and 38 accessions were sequenced for the *atpI-atpH* intergenic spacer and *ndhA* intron, respectively. Haplotype discrimination by HRM was calculated using the *C. subternata* samples sequenced for haplotype confirmation, following the same formula as before.

## Phylogeographic analysis of *C. subternata*

The haplotypes detected via HRM clustering and confirmed by sequencing (described above) were assembled following the same procedure described under 'Developing *Cyclopia* specific HRM primers'. All wild *C. subternata* samples that underwent HRM analysis were then assigned the haplotype identity of the HRM cluster they belonged using a custom *R* script written by A.J.P (provided as File S1 which includes example files for running the script). The cpDNA regions under investigation (*atpI-atpH* intergenic spacer and *ndhA* intron) are maternally inherited in tandem and not subject to recombination (*Reboud & Zeyl, 1994*), and were therefore concatenated for subsequent analysis.

The genealogical relationships among the concatenated haplotypes were determined from a Statistical Parsimony (SP) network (Fig. 5 inset) constructed in TCS *[v1.2.1]* (*Clement, Posada & Crandall, 2000*). Two *C. intermedia* E. Mey. individuals with existing sequence data for the *atpI-atpH* intergenic spacer and *ndhA* intron generated during the primer development phase of the study were included as outgroup taxa. Default options were used to build the network and all indels were reduced to single base-pairs as the software treats a multiple base pair gap as multiple mutations. Haplotype distributions were mapped (Fig. 5) in QGIS *[v3.2.2]* (*QGIS Development Team, 2018*).

The following population genetic differentiation measures were calculated from the concatenated haplotypes: pairwise Gst (*Nei, 1973*), G""st (*Hedrick, 2005*) (both indicators of allele fixation) *Jost (2008)*, which measures allelic differentiation between populations, and Prevosi's dist (*Prevosti et al., 1975*) a measure of pairwise population genetic distance that counts gaps as evolutionary events (all gaps were reduced to single base pair events). These measures provide insight into current allele distributions without assuming historical gene flow patterns (*Jost et al., 2018*). Isolation By Distance (IBD) was evaluated among populations testing the correlation between these genetic differentiation measures and pairwise geographic distance using a Mantel test (*Wright, 1943*) with 9,999 permutations, as implemented using the *ade4 [v1.7]* library (*Dray & Dufour, 2007*; *Kamvar, Tabima & Granwald, 2014*) in R *[v3.5.1]* (*R Core Team, 2018*). In order to account for the possibility of non linear population expansion, relationship between population differentiation measures and the natural logarithm of geographic distance was tested following the same approach (*Rousset, 1997*). Finally, genetic differentiation across the mountain ranges that populations were sampled from was tested via an Analysis of Molecular Variance (AMOVA) (*Excoffier, Smouse & Quattro, 1992*). The mountain ranges included in the AMOVA included: the Tsitsikamma (3 populations, 52 samples), Outeniqua east (2 populations, 31 samples), Outeniqua west (2 populations, 35 samples), and Langeberg (1 population, 24 samples) ranges, as described in Table 5.

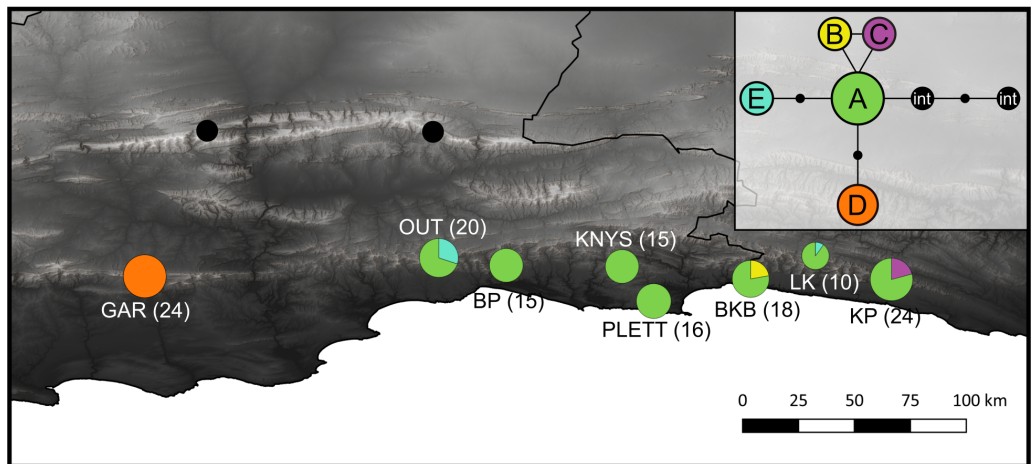

**Figure 5** **Haplotype distribution and number of accessions for the eight *C. subternata* populations screened via HRM.** Black circles mark *C. intermedia* samples collected from the Swartberg mountains and included as out-group taxa. Inset is the genealogical relationship between haplotypes ascertained using the Statistical Parsimony algorithm. Haplotype frequency is indicated as a proportion of the circles representing each population, with total number of accessions provided in parenthesis. The color-coding in the map corresponds to the SP network. Population naming follows the description in Table 4. GAR, Garcia's Pass located in the Langeberg; OUT, Outeniqua Pass and BP, Bergplaas MTO located in the western Outeniqua mountains; KNYS, Diepwalle Knysna and PLETT, Plettenberg Bay in the eastern Outeniqua mountains, and the BKB, Bloukrans Bridge; LK, Langkloof, and KP, Kareedouw Pass in the Tsitsikamma mountains.

**Table 5** **_Cyclopia subternata_ population locations.** The geographic co-ordinates, number of accessions screened via HRM, and haplpotype frequencies (as detected by HRM and verified by sequencing) are given for each *C. subternata* population. Nucleotide differences among haplotypes are provided in Table S3.

| Mountain range | Population | Co-ordinates | | N | Haplotype | | | | |
|---|---|---|---|---|---|---|---|---|---|
| | | S | E | | A | B | C | D | E |
| Langeberg | Garcia's pass (GAR) | −33.96 | 21.22 | 24 | – | – | – | 24 | – |
| Outeniqua W | Outeniqua Pass (OUT) | −33.88 | 22.40 | 20 | 14 | – | – | – | 6 |
| | Bergplaas MTO (BP) | −33.91 | 22.67 | 15 | 15 | – | – | – | – |
| Outeniqua E | Diepwalle, Knysna (KNYS) | −33.92 | 23.14 | 15 | 15 | – | – | – | – |
| | Plettenberg bay (PLETT) | −34.06 | 23.26 | 16 | 16 | – | – | – | – |
| Tsitsikamma | Bloukrans Bridge (BKB) | −33.97 | 23.65 | 18 | 14 | – | 4 | – | – |
| | Langkloof (LK) | −33.87 | 23.91 | 10 | 9 | – | – | – | 1 |
| | Kareedouw pass (KP) | −33.97 | 24.22 | 24 | 19 | 5 | – | – | – |

## RESULTS

### HRM discrimination of sequenced haplotypes

High Resolution Melt curve clustering of haplotypes identified via sequencing for primer development produced variable results: sensitivity ranged from 56%–100%, specificity ranged from 27%–100%, and accuracy ranged from 36%–100% (the number of replicates

**Table 6  Nucleotide varition not differentiated by HRM.**

| Primers | Haplotypes | Nucleotide difference | Specificity |
|---|---|---|---|
| | C-D | T ↔ G & C ↔ A | 18 |
| MLT C1 - MLT C4 (TrnG intron) | B-C | A ↔ C | 20 |
| | B-D | T ↔ G | 29 |
| | A-C | GT ↔ TA, G ↔ T & T ↔ A | 88 |
| | A-D | G ↔ T & T ↔ A | 88 |
| | A-B | GT ↔ TA & G ↔ T | 93 |
| | A-D | T ↔ A | 11 |
| MLT C3 - MLT C4 (TrnG intron) | A-B | 8 bp indel & T ↔ A | 22 |
| | B-D | 8 bp indel | 33 |
| | B-C | T ↔ G & 8 bp indel | 65 |
| | C-D | G ↔ T | 73 |
| | A-C | T ↔ G & T ↔ A | 83 |
| MLT M1 - MLT M2 (pctL-psbE inter-genic spacer) | A-C | G ↔ T & A ↔ G | 6 |
| | B-D | A ↔ G & G ↔ T | 93 |
| MLT S1 - MLT S4 (atpI-atpH intergenic spacer) | A-C | 5 bp indel | 0 |
| MLT V1 - MLT V2 (rpl32-trnL inter-genic spacer) | A-C | 6bp indel, T ↔ A & T ↔ A | 11 |
| | A-D | 6 bp indel & T ↔ A | 93 |
| | C-D | A ↔ T | 96 |

assigned to each HRM cluster is reported in Table 3 and sensitivity, specificity and accuracy is summarized in Fig. 4).

Nucleotide differences between haplotypes failing to produce distinct melt curves, and thus undifferentiated by HRM clustering, are summarized in Table 6. Of the haplotypes not differentiated by HRM: two haplotypes differ by indels, while the remaining 15 comparisons differ by at least one transversion, and two comparisons differed by a transversion and transition. The haplotypes that did produce distinct melt curves differed by at least a transition (26 cases), or multiple SNPs (16 cases), one haplotype differed by a 19 bp indel, and another by a 6 bp indel. All haplotype sequence variation is summarized in Table 3. As previously stated, the three HRM primer combinations with specificity of 100%, two targeting the *atpI-atpH* intergenic spacer (MLT S1–S2, MLT S3–MLT S4) and one targetting *ndhA* intron (MLT U1–U2), were selected for haplotype discovery in wild *C. subternata* populations.

## Detection of haplotype variation in wild populations via HRM

High Resolution Melt curve analysis of accessions from wild *C. subternata* populations revealed no variation in the cpDNA haplotypes amplified by the MLT S3–MLT S4 primer combination, confirmed by sequencing, and the locus was subsequently excluded from further analyses. Five distinct haplotypes were verified by sequencing a subset of samples (ranging from three to eight individuals per population) from each HRM cluster for the remaining two primer combinations.

Of the 142 samples less than 29% were required to be sequenced for haplotype confirmation. Both loci were found to have 100% specificity, i.e., HRM successfully discriminated among all haplotypes detected in wild *C. subternata* populations. However, haplotype richness was overestimated by HRM (sensitivity of 87.6% and 95.5% for MLT S1–MLT S2 and MLT U1–MLT U2 respectively), both cpDNA regions had accuracies of 96%. However, as these additional clusters were sequenced for haplotype confirmation, samples were assigned the true identity of haplotypes resolving any potential issues of low sensitivity.

The final cpDNA dataset comprised 561 bp, 217 bp from the *atpI-atpH* intergenic spacer (MLT S1–MLT S2) and 344 bp from the *ndhA* intron (MLT U1–MLT U2), with a GC content of 29%. An additional 310 base pairs (bp) were amplified by MLT S3–MLT S4, revealing no nucleotide variation. The dataset contained five polymorphic sites; four transversions, one transition, and a seven bp indel (nucleotide differences summarised in Table S3).

### *Cyclopia* subternata phylogeography

The SP network revealed a radiation from a central ancestral haplotype, with few mutations separating haplotypes (Fig. 5 inset). The ancestral haplotype was present in all populations, except the western most Garcia's Pass population, located in the Langberg Mountains. This population contains a single, unique haplotype. An additional two populations (Kareedouw Pass and Bloukrans Bridge) were also found to contain rare, localized haplotypes and a low frequency haplotype was detected in two populations located in the Tsitsikamma and Outeniqua mountains (Fig. 5). Population genetic differentiation measures increased with geographic distance ($R^2 = 0.77$, 0.74, 0.70, and 0.76 for Gst, G"st, Jost's D and Provesti's dist respectively, $p < 0.05$ for all measures), with significance increasing when tested against log transformed geographic distance ($R^2 = 0.64$, 0.67, 0.61, and 0.65 for Gst, G"st, Jost's D and Provesti's dist as before, $p < 0.05$ for all measures). The AMOVA revealed significant ($p < 0.05$) structuring across mountain ranges, accounting for 73.8% of genetic variation (AMOVA results summarised in Table S4).

## DISCUSSION

A nested framework (Fig. 3) was developed to test and apply HRM to non-model organisms, members of the Cape endemic plant genus *Cyclopia*. Polymorphic sites were identified via sequencing 12 non-coding cpDNA regions across 14 *Cyclopia* species. PCR primers for HRM analysis were designed to flank these variable sites, producing 11 HRM primer pairs across 7 regions. Eight of these pairs successfully amplified PCR products and were subsequently analysed via HRM. Specificity of 100% was detected for three of the primer pairs, which were then used to detect haplotype variation in wild *C. subternata* populations with a haplotypes detection accuracy of 96%. Haplotype detection errors were due to false negatives reducing HRM sensitivity. False negatives occur when HRM incorrectly assigns a single haplotype to multiple clustering groups, an issue that is resolved when the haplotype identity of HRM clusters is confirmed by sequencing. Optimized HRM was demonstrated to be a powerful tool for detecting genetic variation in non-model organisms, providing

immediate insights into within population genetic variation via automated melt curve clustering and substantially reduced sequencing efforts. The framework provided here offers a straightforward approach to develop and test the potential application of HRM to non-model systems.

## HRM discrimination of sequenced haplotypes

Differences in DNA melt curves, as detected by HRM, stem from the effects nucleotide sequence chemistry has on melt peak intensity and curve shape. While HRM is reported to be capable of discriminating between any SNP type, the approach may be constrained by physical and chemical properties of the DNA fragment under melt analysis (*Gundry et al., 2008*). Some nucleotide variations, namely class 3 (C ↔ G) and class 4 (A ↔ T) SNPS, tend to produce negligible changes in melt behaviour (curve shape and melt peak) and are often poorly detected by HRM (*Dang et al., 2012*; *Gundry et al., 2008*; *Yamagata et al., 2018*). This is likely to be exaggerated when analysing longer PCR products, as shorter PCR products produce more pronounced melt curve differences than longer nucleotide motifs with the same SNP variation (*Li et al., 2014*; *Liew et al., 2004*; *Taylor et al., 2011*; *Tindall et al., 2009*). Furthermore, nearest neighbour chemistry (the identity of nucleotides directly adjacent to the SNP under investigation) has been shown to impact the melt peak of PCR products, negating any change in melt peak produced by class 3 and 4 SNPs in some cases (*Yamagata et al., 2018*).

Many of these observations are supported by the findings of this study, however some important deviations were detected. Haplotypes that were successfully discriminated by HRM tended to have a class 1 SNP (transitions, C ↔ T and A ↔ G) or multiple SNPs differentiating them. However, seven haplotypes differing by multiple SNPs did not produce distinct melt curves (Table 6), suggesting that some SNPs may potentially counteract one anothers impact on the melt curve. Furthermore, haplotypes that differed by a class 2 (transversions, C ↔ A, G ↔ T) and, as predicted, class 4 SNPs do not appear to have detectable melt curve differences. It is, however, uncertain why in this study some class 2 SNPs produced distinct melt curves in some cases (MLT M1 - MLT M2 and MLT S3 - MLT S4), but not in others (MLT C1-C4 and MLT C3 - C4). Nearest neighbor chemistry does not appear to be provide insights into this as the SNPs had the same neighbouring base pairs across PCR products. Furthermore, a class 2 SNP was differentiated by HRM in a larger PCR product (527 bp) and not in the smaller products (386 bp and 236 bp), indicating that shorter DNA fragments do not necessarily produce more distinct melt curves than larger fragments with the same nucleotide differences.

The primer design choices in this study were largely based on the suggestions that nucleotide variation in shorter DNA strands will have a more pronounced impact on melt curve shape and intensity. This appears to have not been the case and larger PCR products performed as well, if not better, than smaller regions, as detected elsewhere (*Dang et al., 2012*; *Dobrowolski et al., 2009*). Future HRM primer design efforts should possibly explore larger target regions that are more likely to cover multiple SNPs and thus produce more distinct melt curves (*Dang et al., 2012*), such as the products amplified by primer combinations; MLT S1–MLT S2, MLT S3–MLT S4, and MLT U1–MLT U2. This opens

HRM up to exploration of existing universal primers, such as those of *Shaw et al. (2005)* and *Shaw et al. (2007)*, but additional PCR optimization may be required prior to being applied to HRM.

### Detection of haplotype variation in wild Cyclopia populations via HRM

High Resolution Melt analysis using the two best performing primer pairs that amplified variable regions proved to be a highly accurate (96% for both regions screened) means of detecting haplotypes variation in wild *Cyclopia* populations with no cases of different haplotypes occurring in the same cluster (specificity = 100%).

A remarkable feature of HRM is its high and rapid throughput. Running samples in duplicate on a 96 well plate allowed for 48 samples to be screened every three hours. As such, all 142 wild *C. subternata* samples were screened across the two cpDNA regions in two days, with immediate insights into the underlying levels of genetic variation (based on HRM clusterings). This rapid data production comes at a minimal cost per sample, which in this study amounted to $ 11.09 including all PCR amplification and sequencing for the phylogeographic analysis of *C. subternata*. A costing analysis based on quotes obtained in 2017, for a broader *Cyclopia* research project that employed Anchored Hybrid Enrichment (*Lemmon, Emme & Lemmon, 2012*) for nucleotide sequence generation, revealed that, while the cost per bp was not greatly reduced when applying HRM ($ 0.013/bp) as compared to Sanger sequencing ($ 0.015 /bp), and more costly than high throughput sequencing approaches ($ 0.0005 /bp, excluding library preparation and bioinformatic services). The true value of HRM lies in the ability to screen large numbers of samples, with the cost per sample for HRM being 40% that of Sanger sequencing and 16% that of Anchored Hybrid Enrichment.

### Distribution of *C. subternata* genetic diversity

Despite the relatively low genetic differentiation and variation detected across wild *C. subternata* populations, with a widespread haplotype detected in all populations sampled in the Tsitsikamma and Outeniqua mountains, genetic diversity does appear to be spatially structured. Geographically isolated haplotypes were detected in populations in the Tsitsikamma mountains, and complete haplotype turnover was detected in Garcia's Pass population from the Langeberg; possibly a consequence of a genetic bottleneck resulting from a small founding population, facilitating rapid fixation of rare alleles (*Klopfstein, Currat & Excoffier, 2006*). These, and an additional low frequency haplotype shared between Langkloof and Outeniqua populations, provided sufficient divergence across mountain ranges to be detected by an AMOVA and roughly coincide with NJ clustering of populations (Fig. S1). The transition between mountain ranges represents steps of increased genetic differentiation between populations (supported by significant IBD, (*Slatkin, 1993*), and the movement of seed and seedlings across these isolating barriers for Honeybush cultivation should be avoided.

The population divergence described above is in contrast to that reported for the nuclear genome of *C. subternata* (*Niemandt et al., 2018*). While *Niemandt et al. (2018)* also detected a genetically unique population (located in Harlem), no *C. subternata* was detected in this

area during sampling activities despite assistance from landowners in locating wild *C. plicata* Kies populations (iNaturalist observation 14257580) that have been harvested and traded as *C. subternata*. We suggested that additional work be done to describe the *C. plicata* and *C. subternata* populations in this area to confirm potential sympatry between these two morphologically and ecologically similar species (*Schutte, 1997*). No genetic divergence was reported between the two wild *C. subternata* populations (sampled from the Tsitsikamma and Outeniqua mountains) screened and the Agricultural Research Council's (ARC) genebank accessions (*Niemandt et al., 2018*). Genetic material from this genebank has recently been made commercially available for the establishment of cultivated Honeybush stands, including in the Langeberg that supports the genetically distinct GAR population (*Joubert et al., 2011*; *Niemandt et al., 2018*). The effective population size of the *C. subternata* nuclear genome is a scale of magnitude larger than the cpDNA due to the species high ploidy level (hexalpoid, $2n = 6x = 54$, (*Motsa et al., 2018*; *Schutte, 1997*), as such drift may occur more slowly. Additionally, pollen dispersal by carpenter bees (*Xylocopa* spp) may reduce population divergence through rare long distance dispersal events. Seed, in contrast, is dispersed locally by ants (*Schutte, 1997*) and dehiscent seed pods and long distance dispersal is extremely unlikely, unless anthropogenically mediated; this has likely been the case with genetic material actively redistributed across the CFR for the establishment of cultivated populations and breeding trials (*Joubert et al., 2011*).

The geographic distribution of *C. subternata* genetic diversity, as described here, indicates that: (a) unique haplotypes occur within populations, and (b) these unique haplotypes are spatially structured. These patterns of genetic diversity need to be acknowledged in the management of this economically important species, with seed and seedling not translocated outside of the mountain range that they were sourced from.

## CONCLUSIONS

This study demonstrates that HRM is capable of discerning between cpDNA haplotypes, with variable levels of success. When the top performing HRM regions were applied to screening genetic variation in wild populations of the non-model organism, *C. subternata*, all haplotypes were differentiated. While the framework described herein provides a clear guideline on generating the markers required for applying HRM to non-model systems, some analytical adjustments may be required based on the HRM platform available to the lab in question. The high throughput of HRM offers the molecular ecologist the opportunity to increase intrapopulation sample numbers without increasing project costs, while the automated clustering provides real time insights into the underlying levels of genetic variation. Furthermore, this technology may be particularly well suited to the study of conserved and slow mutating nuclear regions and the chloroplast genome of plants (*Schaal et al., 1998*) where low intrapopulation genetic variation is predicted and redundant sequencing of the same nucleotide motifs is likely.

The *Cyclopia* specific primers developed here provide a starting point for assessing potential issues of genetic pollution associated with the transition to commercial Honeybush cultivation (*Potts, 2017*). However, further resolution may be required for

more in depth population studies and additional cpDNA regions as well as low copy nuclear loci should be explored for HRM primer development. Furthermore, the tools produced here, while suitable for phylogeographic work (as demonstrated here), are limited to the maternally inherited chloroplast genome and are not suitable for exploration of interspecific hybrid detection in cultivated Honeybush populations.

## ACKNOWLEDGEMENTS

We would like to thank Gillian McGregor and her students, Nicola van Berkel, Dianne Turner, as well as the Agricultural Research Council of South Africa for their assistance in sample collection and for the valuable discussions on Honeybush ecology and industry. We would also like to thank A. Shutte-Vlock for providing us with samples of hard to collect species and for always being willing to assist in identifying species. We thank the various land-owners and conservation bodies that allowed for sample collection. Additionally, we acknowledge Dr J. R. Alvarado-Bremer and one anonymous reviewer for their comments that improved the quality and clarity of this manuscript.

### Funding

This work was supported by the National Research Fund of South Africa (Grant No. 99034, 95992, 114687) and the Table Mountain Fund (Grant no. TM2499). The funders had no role in study design, data collection and analysis, decision to publish, or preparation of the manuscript.

### Grant Disclosures

The following grant information was disclosed by the authors:
National Research Fund of South Africa: 99034, 95992, 114687.
Table Mountain Fund: TM2499.

### Competing Interests

Alastair J. Potts is an Academic Editor for PeerJ.

### Author Contributions

- Nicholas C. Galuszynski conceived and designed the experiments, performed the experiments, analyzed the data, prepared figures and/or tables, authored or reviewed drafts of the paper, and approved the final draft.
- Alastair J. Potts conceived and designed the experiments, analyzed the data, authored or reviewed drafts of the paper, and approved the final draft.

### Field Study Permissions

The following information was supplied relating to field study approvals (i.e., approving body and any reference numbers):

Sampling was approved by Cape Nature (Permit number: CN35-28-4367), the Eastern Cape Department of Economic Development, Environmental Affairs and Tourism (Permit

numbers: CRO 84/ 16CR, CRO 85/ 16CR), and the Eastern Cape Parks and Tourism Agency (Permit number: RA_0185).

## DNA Deposition

The following information was supplied regarding the deposition of DNA sequences:

The sequences for the non-coding chloroplast regions are available at GenBank: MN879573–MN879581, MN883511– MN883531 and MN930746–MN930802.

## Data Availability

The haplotype clustering results used to determine the accuracy of High Resolution Melt analysis and the sample to haplotype assignment of accessions included in the phylogeographic analysis are uploaded to Figshare and available from the following links respectively:

https://doi.org/10.6084/m9.figshare.11370444.v1,
https://doi.org/10.6084/m9.figshare.11370465.v1.

## Supplemental Information

Supplemental information for this article can be found online at http://dx.doi.org/10.7717/peerj.9187#supplemental-information.

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
