# Peer review of "Application of High Resolution Melt analysis (HRM) for screening haplotype variation in a non-model plant genus: Cyclopia (Honeybush)"

_PeerJ, doi:10.7717/peerj.9187_

## Round 0.1 · original submission · Major Revisions

Dear Dr. Galuszynski,

The reviewers find your study interesting, timely and relevant.

Nevertheless, they have some major concerns that have to be addressed. They also detected several typos, omissions, inconsistencies, and mislabelled figures that need to be corrected.

I encourage you to improve the manuscript according to tips of reviewers. Please, respond point-to-point to the comments of reviewers to speed up the process of revision.

Once again, thank you for submitting your manuscript to PeerJ and we look forward to receiving your revision.

Sincerely,
Gabriele Casazza

·

Basic reporting

This study presents the development of an approach based upon HRM to screen genetic variation contained in chloroplast DNA (cpDNA) of the non-model plant (Cyclopia subternata), endemic to South Africa (aim 1). Sequence analysis was used to develop a battery of genus-specific primers that target 12 non-coding cpDNA regions. The second aim was to test whether HRM could identify the same haplotypes characterized by Sanger sequencing. To that end, four cpDNA segments were characterized using a battery of eight primer-pair combinations. The authors tested these HRM assays for specificity, sensitivity, and accuracy, and concluded that only segment ATPI-ATPH-IS, targeted with primers MLT S1-S2, displayed 100% accuracy, sensitivity, and specificity when identifying four (A-D) haplotypes. A second segment (NdhAx1-NdhAx2) revealed a fifth haplotype (E), and together, the two HRM assays were used to determine the frequency of the five haplotypes by sampling location (Table 4). However, variation in other cpDNA segments targeted with six HRM assays reveal the presence of seven haplotypes (Table 2 and Fig. 5 inset -incorrectly identified in the text as Fig 4), but those HRM's were not used because of they lacked sufficient specificity, sensitivity or accuracy (although they are included in the discussion). The third aim of this article is to provide a case study to use HRM as a high-throughput alternative to identify cpDNA haplotypes for phylogeographic studies of non-model species, using C. subternata as an example. While for the most part this can be considered a seminal study and a guideline for plants, it is not the first to attempt to advance HRM as an alternative for SNP discovery and genotyping in non-model species of animals (see Smith et al. 2013; Marine Genomics 9:39-49).

In its majority, the article is well-written, although it contains several typos, omissions, inconsistencies, and mislabeled figures that need to be corrected (see attached commented PDF). Among the omissions listed in the commented copy, there are a few references that would be appropriate to include.

In addition to the changes included in the commented copy, the following changes are strongly recommended to both figures and tables, and corresponding legends.

Fig. 2. Legend. Be consistent throughout in regard to cpDNA segments or loci analyzed. Reference to primer pairs as to denote regions of cpDNA have very little meaning to the reader and cause a constant distraction by having to go back to tables, and even there, then, the usage of acronyms to refer to the segments is not consistent. Throughout the entire manuscript and tables please refer to the loci or segments by their names, followed by primer pair [e.g., atpI-atpH IS (S1-S2)].

Fig. 4 legend. Substitute regions with loci or segments of cpDNA.
Fig. 4. Modify the x-axis to include the names of the cpDNA segments and corresponding primers. Again, the primer names have very little meaning to the reader, except the authors who are familiar with them and their location.

Fig 5. Legend. Mention the inset (SP network). Also, mention that color-coding in the map corresponds to distinct haplotypes in SP network.
Fig. 5. Inset. Two haplotypes in black, are not commented on the text or figure legend. How were these derived? I presume from sequence data. If so, specify in text.
Fig. 5. Locality names of black circles, not given. Suggest changing the color-scheme used as it is difficult to differentiate the tones of blue.

Table 1. Details for primer MLT_S4 not given. Please check names for consistency, throughout. Primers MLT_V1 and V2, appear elsewhere as R1 and R2

Table 2. Similar to comments of Fig. 2 Legend (above), introduce (group?) names of segments/loci by their names (e.g., atpI-atpH IS) to facilitate the placement of the targetted cpDNA loci subject to HRM. Also, sort the order of segment/primers alphabetically. I have a very difficult time understanding the meaning of columns 1-11 (number of times assigned to haplotype cluster). This is not properly explained neither in methods or in the figure legend.
Again, be consistent with names. Primer names should be the same throughout. MLT S1-S4, not S1-4.

Table 4. Data for Tsitzikamma not included.

Table 5. Legend. Correct spelling.
Table 5. Similar to comments above regarding loci names. Please include.
I

Experimental design

This study represents original research in environmental sciences. Largely the experimental design is appropriate and addresses each of the three aims appropriately. The major concern that I have is with the flowchart of work to characterize the five main haplotypes (A-E). Accordingly, they characterize all the samples with two HRM assays, one with primers S1-S2 (atpi-atpH IS) to obtain the identities of haplotypes A-D. The second HRM using primers U1-U2 (ndh intron) helps identify haplotype E. Since MLT S1-S2 unambiguously determines the identity of haplotypes A-D, and given that the relationship of haplotypes is known through statistical parsimony (SP), then the second HRM should only be applied to individuals belonging haplotype A. Although in this particular example a phylogenetically informative sequential HRM would be of little value as a diagnostic tool (given that Haplotype A dominates most samples), as a generalization for other studies this could be a valuable contribution.

Another area of concern is whether BioRad's High Precision Melt Software normalizes the fluorescent signal, a function of the PCR yield, among samples. Inspection of the left panel of Fig. 2 (HRM) curves suggest that the curves are not normalized, such that some of the differences (and also noise) could result from differential amplification as a function of the quality of DNA of the sample (i.e., sample effect), or as the result that some haplotypes are preferentially amplified over others. Melting curve normalization would remove that noise. prior to calculating the melting curve differences. Further, panel E (also Fig. 2) suggests an incorrect placement in the calculation of the HRM differences, as the amplicon appears to contain two melting domains, with the information of the first compromised by selectin an area still subject to changes in temperature (76-77 degrees), instead of above 84 degrees where all products have been completed melted.

Validity of the findings

Another area of concern regarding the generalization of their approach is the reliance on the automated clustering algorithm of the High Precision Melt Software (BioRad). Not all scientists will utilize the same RT-PCR instrument and software platform and in my experience with four instruments of different brands (HRM-1, Idaho Tech; LC480 & LC96, Roche, and MyGo Mini, IT-IS-LifeSciencies) the reliability to determine similar melting curves vary, and the clustering algorithms can become subjective. Accordingly, the authors should modify lines 437-438 HRM to reflect that the guidelines provided therein to generate markers in non-model species may need to be adjusted according to the clustering algorithm provided by the RT-PCR manufacturer.

Reviewer 2 ·

Basic reporting

Please check some typos along the text
Lines 42, 90, 136, 153, 172, 189, 190

Experimental design

Overall, the results are interesting and supported by a good experimental design.

Validity of the findings

No comment

Additional comments

My main concern is rather a general comment: it is not clear if the aim of the MS is more methodologic or descriptive. I've found the Material and Methods too long. In Material and Methods, a number of generic information has been described and some methodological results have been discussed. In my opinion, authors should discuss some information provided in M&M such as results and emphasize the primers selection work.

Reviewer 3 ·

Basic reporting

1. The article is well written and it contains valuable novel research findings in a field where limited information on Cyclopia species is available. These research findings are contributing meaningful knowledge on population diversity and therefore should be published.
2. The reference to supplemental Tables, Figures exc. in the article, which is not always available to readers, should rather be included in the article. It will make the article more informative and give a better understanding of the scope of the study.
3. In line 109 the authors refer to the 14 species used in the study as Table 1, but is actually Table S1. It is recommended that this Table is included as part of the article, (Table 1) since the 14 species are not mentioned anywhere else and it is valuable information to readers and scientists. All other Tables numbering need therefor be changed accordingly.
4. In line 406 Fig S1 with the population tree is very informative and be included as part of article.
5. Other reference to supplemental material can be omitted e.g. Table S4 in line 388.
6. None of the supplemental Tables, Figures etc. had titles to inform reader what it is about.
7. The supplemental fasta formatted files could not be opened and verified due to not having the necessary software.
8. The reviewer had not access to verify the permits etc., but they were mentioned in article and it is accepted that they were in order, otherwise collection of wild species would have been impossible.
9. The following references were not in the reference list – Galuszynski & Potts, 2017; van der Bank et al, 2002 (lines 141 and 142)
10. References in line 269 (R Core Team 2018) and line 259 (QGIS Development Team 2018) are also not in reference list. In most Journals they are also required to be in reference list with more information on them or articles published on them.
11. In the abstract under aim - line 6,7 and 8 the word “to” to be added to the aim
a) to develop genus-specific primers for High Resolution Melt analysis (HRM) of members of Cyclopia Vent., b) to test the haplotype discrimination of HRM compared to Sanger sequencing, and c) to provide a case study
12. Spelling errors –
Please note that Bloukrans is spelled with an “s” and not a “z”. Please change it throughout document
Line 43 - among
Line 172 – primers
Line 314 – Kareedouw and not Kareedow
Line 404 – Langkloof and not Langekloof
13. Other
For clarity of high ploidy level mentioned in line 420 either say in brackets after ploidy level (hexaploid) or 2n=6x=54
14. Tables and Figures
• Please make sure all species names etc. in headings and tables are in Italics.
• Please remember Tables and Figures headings should be clear and readers should not need to refer back to document to understand the tables and figures
• Table 1 (should be 2) – reference to Table S2 – is this valid or helpful. TM in Table should be Tm and defined below the Table as melting temperature or heading. Degree missing C at TM in Table.
• Table 3 heading Tm needs to be defined
• Table 4 – availability of Table S3?
• Table 5 – spelling of location
Diepwalle and not Diepwelle
Langkloof and not Langekloof
Kareedouw and not Kareedow
Bloukrans and not Bloukranz
• Figure 1 – Adding a few main towns (Cape Town, Port Elizabeth, George, Barrydale, Uniondale) into the map will make it easier to identify locations of collections. It would have been nice if the collection sites for the different species could have been indicated as well, but I guess it will make the map too messy.
• Figure 5 spelling of locations
Diepwalle and not Diepwelle
Langkloof and not Langekloof
Kareedouw and not Kareedow
Bloukrans and not Bloukranz
15. Results and Discussion – See points at “Validity of the findings”
Authors should also remember ARC only sold seed in the past 7-8 years while cultivation started in 1996. It is still practice of farmers to collect seed in the wild or get from their neighbours and are probably more responsible in spreading plant material to new areas. However, in both cases it should be monitored, especially areas with unique populations.
16. Acknowledgements
The Agriculture Research Council (ARC) is not acknowledge for making their genebank accessions available to the study. Organisations allowing the authors to collect plant material on their land e.g. SANPARKS, farmers etc. should also be acknowledged. Normally funders are also thanked or acknowledged here unless there is a specific heading for funding.

Experimental design

1. The material and methods are clearly described and in length described how primers are developed and chosen.
2. A wide area was covered and sampled and several individuals in a population were included in the dataset. The collection of plant material alone was a huge and tiresome task. Therefor making the data more reliable and useful.
3. Again, the availability of supplemental Tables, Figures, files are questionable.
4. It would have been good to know in which areas the 14 species and their populations were collected, but is not necessarily adding value to this article.

Validity of the findings

1. In line 413 the authors made a personal observation stating that the unique population found by Niemandt et al (2018) is C. plicata and not C. subternata. They made no explanation on how they got to that conclusion. Niemandt et al (2018) stated clearly that samples of this population were identified by botanists at SANBI. Furthermore, the leaf shape of these two species are distinct (needle like vs oblong) and the two species cannot be wrongly identified as the other. Locals at Haarlem also identified the population as C. subternata, who are familiar with the species occurring in that area. In Niemandt et al study the SSR markers identified and grouped with these of C. subternata and not with any other species in the study. The chromosome number is also the same as C. subternata, although the chromosome number for C. plicata is still unknown. Both species do occur in the area of collection, but the population collected by the current authors are a few kilometers away from Niemandts sample area. Therefore, authors should omit this statement without any proof. They can rather expand on the phenomenon of unique populations in the Haarlem area found in two studies. It can be speculated that this is perhaps subspeciation and that more studies are required on populations in this area. Definitely something interesting to follow up
2. Did the authors used the ARC gene bank accessions of C. subternata (breeding population) as a group to compare how they vary from the C. subternata wild populations. This will be a good indication if contamination had took place or may take place. If the breeding population are sharing the same genetic material as the wild populations in an area, the risk for genetic contamination or “pollution” is minimized. This will be a valuable exercise to do and helpful to the honeybush industry.

Additional comments

1. The reviewer wants to congratulate the authors with the study and their contribution to science and knowledge on honeybush, and their initiative to use new methodologies to obtain and create new data and understanding of this genus. Honeybush is a valuable niche crop and the more and the better we understand the crop, additional benefit and growth can be added to the industry.
2. It is with concern that it was noticed the authors uploaded this unreviewed article to an open access website, Researchgate, before the article was accepted. They also didn’t upload the supportive data, Tables etc. that are referred to in the article, thus making the information less usefull e.g. the 14 species used in the study. It is recommended that authors only upload articles to websites after articles are accepted and if it is in accordance to the Journals policies.

---

## Round 0.2 · Minor Revisions

Dear Dr. Galuszynski,

The reviewers find only some minor editorial errors that need to be fixed. So, I ask you to perform these few changes before the manuscript is accepted for publication

Once again, thank you for submitting your manuscript to PeerJ and we look forward to receiving your revision.

Sincerely,
Gabriele Casazza

·

Basic reporting

Line 192. Delete extra period after 'analysis.'
Line 201. Capitalize 't' to upper case in 'table 3'
Line 632. Correct font

Table 3. The right portion of the table does not fit, and thus cannot be read.

Experimental design

This version represents a substantial improvement to all the deficiencies noted in the description of the experimental design and methods.

Validity of the findings

Findings are valid and sound with important ramifications for rapid genotyping and biogeography of non-model plant species.

Additional comments

I am completely satisfied with the manner you addressed that suggestions made in my original review, as well as the way you treated the remarks of the other two reviewers, in particular reviewer number 3.

Reviewer 3 ·

Basic reporting

1. Still several errors with the names of localities, towns etc. Tsitsikamma is without a z.
Please make sure all spelling is correct as it appears in roadmaps, google maps etc.
2. Figure 1 is now much more informative
3. Please check that all genus and species names are in Italics, especially in Table and
Figure headings , and spelling of locations
a. Figure 1 (Cyclopia and C. subternata) Italics
b. Figure 5 Bergplaas and Tsitsikamma
c. Table 3 not Landscape
d. Table 5 Heading – change “ of each Cyclopia subternata populations” to “are
given/listed for each Cyclopia subternata population”. It does not read correctly.
e. In Table 5 – spelling Tsitsikamma
4. Typing corrections in
a. Line 185 – delete extra “.”
b. Line 189 – capital T in table 3
c. Line 286 – Tsitsikamma with s and not Z
d. Line 360 – Delete space between SNPS and ,
e. Line 566 - Delete ?
f. Line 623-625 – Change Font

Experimental design

No comment

Validity of the findings

1. Agree that Haarlem population may need re-evaluation and further study. Not sure if samples were taken in same area as current authors. However, botanist at SANBI is well familiar with Cyclopia species that was part of his PhD studies, using flowering samples for identification and compared it with flowering C. subternata. As mentioned by authors due to environmental changes and stresses species can be misidentified. Schutte 95, 97 mentioned several varieties and includes C. latifolia in her classification of C. subternata. Did Schutte or any botanist verified all populations used in this study?
Paragraph reads better and can be accepted.
2. It’s a pity the authors lost the ARC genetic material, which could have helped to clarify and support the research findings in this study.
3. Not familiar with any cultivation activities in the Garcia area.

Additional comments

No comment

---

## Round 0.3 · accepted · Accept

Dear Dr. Galuszynski,

I am very pleased to say that your paper "Application of High Resolution Melt analysis (HRM) for screening haplotype variation in non-model plants: a case study of Honeybush (Cyclopia Vent.)" is accepted for publication in the PeerJ. Congratulations!
Thank you for submitting your work to PeerJ.
Sincerely,
Gabriele Casazza